# Copper-Fructose Interactions: A Novel Mechanism in the Pathogenesis of NAFLD

**DOI:** 10.3390/nu10111815

**Published:** 2018-11-21

**Authors:** Ming Song, Miriam B. Vos, Craig J. McClain

**Affiliations:** 1Department of Medicine, Division of Gastroenterology, Hepatology and Nutrition, University of Louisville School of Medicine, Louisville, KY 40202, USA; cjmccl01@louisville.edu; 2Hepatobiology&Toxicology Center, University of Louisville School of Medicine, Louisville, KY 40202, USA; 3Department of Pediatrics, Emory University School of Medicine, Atlanta, GA 30307, USA; MVOS@emory.edu; 4Children’s Healthcare of Atlanta, Atlanta, GA 30322, USA; 5Department of Pharmacology and Toxicology, University of Louisville School of Medicine, Louisville, KY 40202, USA; 6University of Louisville Alcohol Research Center, University of Louisville School of Medicine, Louisville, KY 40202, USA; 7Robley Rex Veterans Affairs Medical Center, Louisville, KY 40206, USA

**Keywords:** copper, fructose, kupffer cell (KC), iron, non-alcoholic fatty liver disease (NAFLD), metabolic syndrome, gut microbiota

## Abstract

Compelling epidemiologic data support the critical role of dietary fructose in the epidemic of obesity, metabolic syndrome and nonalcoholic fatty liver disease (NAFLD). The metabolic effects of fructose on the development of metabolic syndrome and NAFLD are not completely understood. High fructose intake impairs copper status, and copper-fructose interactions have been well documented in rats. Altered copper-fructose metabolism leads to exacerbated experimental metabolic syndrome and NAFLD. A growing body of evidence has demonstrated that copper levels are low in NAFLD patients. Moreover, hepatic and serum copper levels are inversely correlated with the severity of NAFLD. Thus, high fructose consumption and low copper availability are considered two important risk factors in NAFLD. However, the causal effect of copper-fructose interactions as well as the effects of fructose intake on copper status remain to be evaluated in humans. The aim of this review is to summarize the role of copper-fructose interactions in the pathogenesis of the metabolic syndrome and discuss the potential underlying mechanisms. This review will shed light on the role of copper homeostasis and high fructose intake and point to copper-fructose interactions as novel mechanisms in the fructose induced NAFLD.

## 1. Introduction

Accumulating evidence has shown that increased fructose consumption parallels the rises in the obesity epidemic, metabolic syndrome and NAFLD in the United States and worldwide [1,2,3,4,5,6,7,8]. Moreover, fructose consumption is higher in patients with NAFLD compared to healthy controls and is associated with severity of fibrosis, suggesting that high fructose intake may be an important risk factor for the development and progression of NAFLD [9,10,11,12]. Fructose is consumed mainly as added sugars, such as sucrose and high-fructose corn syrup (HFCS), which represents 45% and 41% of the total added sugars ingested, respectively [13].

Fructose is distinct from glucose due to its unique metabolism [14] and limited absorption [15,16]. The role of fructose in the induction of components of metabolic syndrome, as well as NAFLD, has been well documented in numerous animal studies [17,18,19,20]. A recent study demonstrated dietary fructose is primarily metabolized in the small intestine. However, excess fructose ingestion can saturate intestinal clearance capacity; it then reaches the liver and colon microbiota where it is metabolized [21]. In line with these data, depletion or knockdown of ketohexokinase (KHK), a key enzyme of fructose metabolism, markedly attenuated high fructose diet-induced NAFLD, obesity and other metabolic effects [22,23,24]. Similarly, toll like receptor 4 (TLR4) mutation or oral antibiotics protected against high fructose diet-induced NAFLD in mice [17,25], highlighting the importance of the gut-liver axis in the pathogenesis of dietary fructose associated NAFLD and metabolic syndrome. Despite the major progress that has been made over the past two decades, the mechanisms underlying fructose-induced NAFLD and metabolic syndrome are still incompletely understood. Even with increased *de novo* lipogenesis, only a small amount of fructose (<1%) ingested is converted to plasma triglyceride [26]. Thus, it was proposed that the general activation of lipogenesis and blockade of fatty acid oxidation signaling might account for the fructose induced fat accumulation in the liver [27]. However, a gap still remains in our understanding of increased *de novo* lipogenesis and hepatic fat accumulation during ingestion of fructose.

Although compelling epidemiologic data support the critical role of dietary fructose in the epidemic of the metabolic syndrome [2,8,28,29,30,31], a causal link between fructose consumption and the metabolic syndrome has not been firmly established in human studies [32,33,34,35,36]. Most of studies on fructose are limited by a short term of study and/or a small study population. Moreover, hypercaloric effects cannot be excluded in some of the studies [37]. Of note, isocaloric dietary fructose restriction has been reported to be beneficial in improving obesity and metabolic parameters [38,39,40]. Animal studies also showed the essential role of fructose in the methionine-choline-deficient (MCD) diet-induced nonalcoholic steatohepatitis (NASH) model [41,42]. Thus, another key issue is the complexity of the etiology of NAFLD, which involves multiple metabolic effects. One crucial factor is the nutrition interactions. Results from animal studies demonstrated that when fructose is ingested with fat, more severe hepatic steatosis and liver injury or fibrosis was induced compared to high fat diet alone, even when consumed isocalorically [43]. Similarly, the metabolic effects of fructose are more obvious in obese patients with NAFLD and insulin resistance [12,32,44,45,46], suggesting potential interactions between fructose and other metabolites or a complex interplay with other metabolic pathways. We and others have demonstrated copper-fructose interactions in inducing the components of metabolic syndrome and NAFLD in rat model [47,48,49,50,51].

A growing body of evidence indicates that hepatic copper level is lower in NAFLD patients, and steatosis grades inversely correlate with hepatic copper content [52,53,54,55,56]. Moreover, dietary copper restriction induces hepatic steatosis and insulin resistance in rats, suggesting that copper availability may be involved in the development of NAFLD [52]. The mechanism leading to low copper levels in NAFLD patients is not clear. Multiple factors can lead to copper deficiency, including the amount of copper in the diet. The Western diet often is low in copper [57,58]. Other factors, including bariatric surgery, pancreatoduodenectomy, excessive use of denture cream high in zinc and excessive intake of soft drinks, with added fructose, can also induce copper deficiency by impairing copper absorption [59,60,61,62,63,64,65,66,67].

Collectively, high fructose consumption and inadequate copper intake represent two important nutritional problems in the United States. Although copper-fructose interactions have been well documented in experimental models [48,49,50,51,68,69,70,71,72], limited data are available from human studies. In this review, we will discuss the role of copper-fructose interactions in the pathogenesis of the metabolic syndrome and NAFLD and discuss potential underlying mechanisms.

## 2. Epidemiology of NAFLD, Fructose Consumption, and Dietary Copper Intake

NAFLD is now the most common liver disease in the United States, and accounts for more than 75% of chronic liver diseases. In contrast with the other chronic liver diseases, whose prevalence has remained stable, the prevalence of NAFLD increased steadily from 5.51% (1988–1994) to 9.84% (1999–2004) to 11.01% (2005–2008) [73]. The most recent study showed that global prevalence of NAFLD is 25.24%, and is 24.13% in North America [8]. The increased prevalence of NAFLD parallels the increases in obesity, type 2 diabetes (T2D), insulin resistance and hypertension [8,73]—all hallmarks of the metabolic syndrome. In particular, the prevalence of suspected NAFLD in adolescents increased at an alarming rate, from 3.9% to 10.7% over the past 20 years [7]. NASH, a more advanced stage of NAFLD, is the third most common indication for liver transplantation in the United States [74]. Of note, NAFLD and NASH exhibit age and sex differences, with a higher prevalence in males than in females in both adolescents and adults until the age of 60. After age 60, the prevalence of NASH is higher in women [75,76]. Based on NHANES III (1988–1994) data, mean fructose consumption was 54.7 g per day and accounted for 10.2% of total caloric intake [4], about 50% higher than the mean reported from the 1970’s (37g/day) [5]. Consumption was highest among adolescents at 72.8 g/day (12.1 % of total calories) [4]. Although evidence showed that consumption of sugar-sweetened beverages (SSBs), which are the leading source of added sugars in the diet, has declined recently [77,78], it remains high among children and adolescents [79]. Moreover, time-trend data over the past 3 decades have shown that the increased consumption of SSBs parallels the obesity epidemic and is associated with increased T2D risk in the United States [2]. The prevalence of obesity increased from approximately 5% (early 1970’s) to 17% (2011–2014) in children and adolescents, and from 15% (late 1970’s) to 36.5% (2011–2014) in adults in the United States [80]. The rise of SSBs intake is mainly due to the dramatic increased consumption of HFCS, which is the primary sweetener in SSBs [2,13]. The most common forms of HFCS contain either 42% (HFCS-42) or 55% (HFCS-55) fructose, along with glucose and water, with HFCS-55 being the most common used form [13,81]. Therefore, SSBs appear to be the major source of dietary fructose. It was estimated that 184,000 global deaths in 2010 were attributable to consumption of sugary beverages, with 72.3% from diabetes mellitus, 24.2% from cardiovascular diseases (CVDs) and 3.5% from cancer. United States is ranked second in SSB-related mortality among the 20 most populous countries in the world [82]. Accordingly, Dietary Guidelines for Americans recommends decreasing added sugars from 25% (2010-2015) to less than 10% (2015–2020) of calories per day [31,83].

The Recommended Dietary Allowance (RDA) for copper in adult men and women is 0.9 mg/day and the Estimated Average Requirement (EAR) for copper is 0.7 mg /day [84]. Klevay summarized the data from NHANES III and found that at least one fourth of adults consume less than the EAR in both the United States and Canada [59,85]. A recent study revealed that 62% and 36% of diets of 80 randomly selected adults in Baltimore were below the RDA and EAR, respectively [86], suggesting that the Western diet is often low in copper. Secondary copper deficiency can be caused by factors such as gastric bypass surgery and high zinc exposure [87,88].

Copper status is affected by age, gender and hormone use. Plasma copper concentrations and ceruloplasmin levels were higher in women than in men [89]. Lack of good biomarkers make it challenging to monitor marginal copper deficiency status [88]. The copper-containing enzyme activities in blood cells, such as erythrocyte copper/zinc-superoxide dismutase (SOD1) and platelet or leukocyte cytochrome c oxidase (COX), are sensitive to changes in copper stores and are better indicators of copper status than plasma copper level and ceruloplasmin activity [89,90]. A recent study by Heffern et al. revealed that Copper-Caged Luciferin-1(CCL-1), a bioluminescent reporter, can be used for tissue-specific copper in vivo imaging, thus to monitor copper levels in living animals. They found that hepatic copper levels were markedly decreased in diet-induced NAFLD mice model [91].

## 3. Fructose Absorption, Metabolism and Metabolic Fate

Dietary fructose is absorbed in the small intestine, and the absorption of fructose is greater in the proximal and middle than in the distal small intestine [92]. A murine study with oral gavage of low dose ^13^C-fructose demonstrated that labeled fructose 1-phosphate (F1P), a specific metabolite of fructose, predominantly localized in the small intestine (jejunum ˃ duodenum ˃ ileum). When high dose of ^13^C-fructose was orally gavaged to mice, the majority of labeled F1P was detected in the jejunum and only a small amount of labeled F1P accumulated in the liver, suggesting that most of ingested fructose is metabolized in the small intestine and only excess fructose intake flows to the liver [21]. Fructose is absorbed into enterocytes by the fructose transporter GLUT5 in the apical membrane and exits to the portal blood via GLUT2 in the basolateral membrane of enterocytes. The absorption of fructose in the liver is mediated by GLUT2 [93]. Recent studies demonstrated that GLUT8 also play an important role in the hepatic and intestinal fructose absorption. Moreover, GLUT8 mediated fructose absorption exhibits sex differences [94,95,96]. Of note, the distribution of GLUT2 and GLUT8 in the liver as shown by mRNA abundance is very similar between humans and mice [95]. GLUT5 is mainly expressed in the small intestine and kidney, while the major sites of GLUT2 expression are the liver, pancreas, intestine, kidney, and brain. The distribution of GLUT5 in the small intestine exhibits a regional pattern which is greater in the proximal segment compared to the distal segment [97]. Moreover, GLUT5 is inducible and dramatically stimulated by early introduction of dietary fructose [97]. *Glut5* deletion resulted in more than a 75% reduction in fructose absorption in the small intestine and a decrease of 90% of serum fructose concentration compared to wild-type mice [98]. In addition, GLUT5 gene expression is tightly regulated by glucocorticoid and thyroid hormones and circadian rhythm [97,99].

Fructose is metabolized by KHK or fructokinase. [23,100,101]. Fructose is phosphorylated by KHK at the 1-position to generate F1P, which consumes ATP and phosphate. F1P is then cleaved to glyceraldehyde and dihydroxyacetone phosphate (DHAP) by aldolase B. At this point, glucose metabolism and fructose metabolism converge. Unlike glycolysis, fructolysis bypasses phosphofructokinase, a rate-limiting step in glycolysis, to circumvent feedback inhibition. Moreover, KHK is much faster than hexokinase in phosphorylating their substrates [14], thus leading to rapid ATP depletion and phosphate consumption [102]. There are two KHK isoforms, KHK-C and -A, and both can metabolize fructose, but KHK-C is considered the primary enzyme involved in fructose metabolism because of its lower Michaelis constant (Km) [23,101]. Depletion of KHK-C or KHK-A and -C, but not KHK-A alone, protects against fructose-induced metabolic syndrome [23,103,104]. A reduction of intracellular phosphate leads to the activation of adenosine monophosphate deaminase (AMPD), which converts AMP to inosine monophosphate (IMP). IMP is subsequently converted to hypoxanthine and then to xanthine, ultimately leading to the generation of uric acid [105,106,107] (Figure 1). Inhibition of xanthine oxidase (XO), a rate-limiting enzyme that catalyzes uric acid production, protects against hepatic steatosis in mice [108].

The metabolic fate of fructose has been shown by studies with an isotope tracer. After ingestion of fructose, approximately 50% is converted into glucose, 25% is converted into lactate, 17% is converted to glycogen, and only less than 1% is converted to plasma triglyceride. However, most of the tracer studies are short-term studies. Longer term effects of fructose on the *de novo* lipogenesis need to be evaluated [13,26]. It appears that fructose-induced fatty liver is unlikely the direct effect of fructose metabolism. This leads to the hypothesis that the activation of lipogenesis and blockade of fatty acid oxidation signaling might account for the hepatic steatosis induced by fructose metabolism [27]. Overall, knowing the fundamentals of fructose biochemistry is crucial for the understanding of fructose induced metabolic disorders.

Recently, work from animal studies demonstrated that endogenous fructose, generated from polyol pathway, plays a critical role in the development of metabolic syndrome and NASH (in addition to dietary fructose) [22,104]. The polyol pathway is an alternate route of glucose metabolism. The rate-limiting step of this polyol pathway is the reduction of glucose to sorbitol which is catalyzed by aldose reductase (AR). Under normoglycemia, AR-catalyzed reduction is less than 3% of total glucose utilization, whereas more than 30% glucose is used by AR under hyperglycemia [109,110].

The causal role of fructose in the pathogenesis of NAFLD has been demonstrated in numerous animal studies [17,25,111,112] and has been reviewed [113,114,115,116]. In this review, we will focus on the copper-fructose interactions and NAFLD.

## 4. Copper Absorption, Distribution and Utilization

Copper is an essential trace element. It serves as a cofactor for a number of enzymes, including COX, SOD1, ceruloplasmin, hephaestin, and lysyl oxidase, etc., which are involved in mitochondrial respiration, antioxidant defense, copper and iron export, connective tissue maturation, etc. [117]. In addition, copper also functions as a cellular signal to regulate cellular and molecular events, such as proteasome degradation of copper chaperone for SOD1 (CCS) and hypoxia inducible factor-1 (HIF-1) activation [118,119]. Mammals acquire copper through the diet. Copper absorption, distribution, and utilization are tightly regulated to maintain copper homeostasis. Dietary copper is primarily absorbed from the small intestine via copper transporter 1 (Ctr1). Ctr1 is considered the major copper transporter in mammalian cells [120,121,122]. Mice with intestinal-specific deletion of Ctr1 died of severe weight loss within three weeks, and these mice can be rescued by a single intraperitoneal injection of copper [123]. Similarly, cardiac-specific knockout of the Ctr1 results in cardiac copper deficiency and severe cardiomyopathy [124], suggesting that Ctr1 is required for copper absorption. Ctr2 was originally identified based on its sequence homology to Ctr1. However, Ctr1 and Ctr2 exhibit distinct functions in copper metabolism. Ctr2 knockout mice exhibit increased tissue copper levels and are defective in accumulation of truncated Ctr1. Thus, Ctr2 plays a significant role in Ctr1 degradation and functions as a regulator of Ctr1 [125,126]. In human adults, the amount of copper absorption is inversely correlated with dietary copper intake; high dietary copper intake results in low copper absorption [127]. After import, the copper ion in the cytoplasm is stored either in a complex with metallothioneins (MT) mediated by GSH, or is distributed to proteins or organelles by specific Cu chaperone proteins that function in the delivery of Cu to mitochondrial COX (via Cox17), to SOD1 (via CCS), and to the cytosolic Cu binding domain of the P-type Cu-transporting ATPases, ATP7A or ATP7B (via Atox1) [128]. ATP7A and ATP7B are required for transport of copper into the trans-Golgi network (TGN) for biosynthesis of several secreted cuproenzymes and for efflux of copper. ATP7A is required for copper efflux in the intestine and ATP7B is required for the biliary excretion of excess copper in the liver [128] (Figure 2).

## 5. Copper Homeostasis and NAFLD

Disturbance of copper homeostasis is associated with a variety of clinical manifestations. In this review, we focus on the copper dyshomeostasis associated NAFLD and its risk factors.

Analysis of 124 adult biopsy-proven NAFLD patients revealed that serum copper as well as liver copper levels are lower compared to healthy controls and patients with other types of liver diseases, including hepatitis C virus (HCV) infection, autoimmune hepatitis, and alcoholic liver disease. Among these NAFLD patients, NASH patients displayed even lower hepatic copper levels than those with simple steatosis. Hepatic copper level is lower in NAFLD patients with the metabolic syndrome and T2D compared to those without metabolic syndrome and T2D [52]. Moreover, NAFLD patients with lower serum copper and lower liver copper exhibited higher serum ferritin levels and hepatic iron levels, which were associated with decreased mRNA expression of liver ferroportin-1 (FP-1) [53]. Similar results were obtained from pediatric NAFLD patients [54,56]. More severe NAFLD (NAFLD activity score, NAS ≥5) patients, particularly in those with ballooning hepatocytes, displayed significantly lower serum copper and ceruloplasmin levels compared to the patients with less severe NAFLD (NAS ˂ 5) [56]. A recent study from 751 Korean adults revealed that lower hair copper concentration was associated with higher body mass index, waist circumference, blood pressure, and lower levels of high-density lipoprotein cholesterol. Of note, NAFLD patients displayed significantly lower hair copper concentrations [129]. Moreover, dietary copper restriction induces hepatic steatosis and insulin resistance in rats, further suggesting that copper availability may be involved in the development of NAFLD [52].

Mutations in the ATP7B gene leads to Wilson’s disease (WD), an inherited autosomal recessive disorder of copper dyshomeostasis, characterized by excessive hepatic copper accumulation and decreased serum ceruloplasmin levels. In the earlier stage, it manifests as hepatic steatosis which is often indistinguishable from NAFLD [130]; it may progress to hepatic fibrosis and cirrhosis, and eventually liver failure [131]. The mechanism(s) by which hepatic copper accumulation leads to hepatic steatosis are not clear, but likely involve mitochondrial damage [132], global DNA hypomethylation [133], and/or nuclear receptors [134]. In addition to WD, evidence from animal studies indicates a critical role of copper homeostasis in the pathogenesis of liver fibrosis. Bile duct ligation results in copper accumulation in the liver. High copper levels were also observed in the late stage of NAFLD patients, including cirrhosis and hepatocellular carcinoma [135]. Treatment with a copper chelator protects against bile duct ligation-induced liver fibrosis. However, overdose of copper chelator results in copper deficiency and accentuates liver injury and fibrosis [136,137]. Thus, both copper deficiency and excess may lead to hepatic steatosis and, in some cases, more severe distinct liver pathology. The relationship between copper and NAFLD has recently been reviewed [138,139].

## 6. Copper-Fructose Interactions.

Extensive studies in 1980’s demonstrated that dietary copper-fructose interactions worsened copper deficiency-induced metabolic syndrome. The severity of experimental copper deficiency was exacerbated by a diet containing high fructose compared to animals with diets containing high glucose or starch [50,51,69,70,71], and this was characterized by lower body weight and hematocrit, and increased liver weight, blood urea nitrogen, ammonia, cholesterol and triglycerides. Switching the type of dietary carbohydrate from fructose to either starch or glucose ameliorated the severity of copper deficiency [50]. In line with animal studies, a human study demonstrated that adult males displayed significantly reduced SOD1 activity in erythrocytes after consumption of a low copper (1.03 mg/day/2850 kcal) and high fructose (20% calorie) diet for 11 weeks compared to those who consumed diets with low copper and starch [60], suggesting dietary fructose intake can affect indices of copper status.

### 6.1. Copper-Fructose Interaction and NAFLD

Our studies demonstrated that dietary high fructose intake further impaired copper status and exacerbated liver injury and fat accumulation in marginally copper deficient rats (Figure 3) [48]. Similar results were obtained in rats fed with high sucrose and copper deficient diet [140]. Moreover, we found that not only high dietary fructose (30% (*w*/*v*) fructose in the drinking water) impairs copper status, but also modest fructose consumption (3% (*w*/*v*) fructose in the drinking water) has a similar adverse effect on copper status [48,49]. The limitation of these studies is the AIN-76 based rodent diet which contains 49% sucrose, which could be a potential confounding factor. However, when extra fructose was given from drinking water, it still worsened the copper status.

Of note, the expression of copper transporter, Ctr-1, in duodenum was markedly upregulated when animals were exposed to a marginal copper deficient diet, and this upregulation was abrogated by high fructose feeding [48], suggesting that high fructose intake may impair copper absorption, which is likely a mechanism underlying copper-fructose interactions. Results from previous studies also support the concept that impaired copper absorption from gut might account for the more severe copper deficiency associated with copper-fructose interaction [61,62]. How dietary fructose impairs copper absorption and whether it is mediated by Ctr-1 remain open questions.

The mechanisms by which copper-fructose interaction induces NAFLD are not clear. Marginal copper deficient diet with high fructose feeding (CuMF) significantly upregulates hepatic fatty acid synthase (FAS) protein expression compared to either marginal copper deficient diet or high fructose feeding alone [48]. Copper-fructose interaction induced hepatic steatosis is completely abrogated by Kupffer cell (KC) depletion, which is associated with the downregulation of hepatic sterol regulatory element-binding protein-1 (SREBP-1) [48,141]. Upregulation of FAS and SREBP-1 by copper deficiency was also observed in other studies [142,143,144]. Pretreatment of KCs isolated from CuMF rats with an intracellular lysosomal iron chelator significantly attenuated lipopolysaccharide (LPS)-induced monocyte chemoattractant protein-1 (MCP-1) production in culture medium, suggesting that the MCP-1 signaling pathway was mediated, at least partially, by intracellular iron (141). A role for MCP-1 in inducing steatosis in hepatocytes has been described [145,146]. The precise mechanism underlying the role of KC in CuMF induced hepatic steatosis remain to be defined.

Iron overload is considered as a partial potential mechanism underlying copper deficiency and fructose induced metabolic syndrome [147,148,149,150]. We showed that marginal copper deficient and high fructose diet markedly increased liver iron level (Figure 4) [49] as well as plasma ferritin level in rats [48]. Similarly, NAFLD patients with low copper levels had hepatic iron overload [52,53]. Mechanism(s) by which copper deficiency induces iron overload have been partially elucidated. Cellular iron export requires members of a family of copper-containing ferroxidases, including ceruloplasmin and hephaestin which oxidize iron from the ferrous to ferric forms. The ferric form of iron binds to Apo-transferrin, thereby facilitating transferrin delivery to peripheral organs. Hephaestin functions to move iron across the basolateral membrane of intestinal epithelial cells into the circulation. Hephaestin-deficient mice display iron deficiency anemia with accumulation of iron in enterocytes [151]. Ceruloplasmin exerts its action on intestinal iron absorption, iron release from macrophages and hepatocytes [152,153]. A clinical phenotype of NAFLD that we regularly see is a young adult male with modestly decreased serum ceruloplasmin, increased serum ferritin, and high fructose intake via sugared pop.

Decreased activities of cuproenzymes, such as SOD1 and COX [154,155], may lead to decreased antioxidant defense and mitochondrial dysfunction, which are likely mechanisms leading to liver injury and hepatic fat accumulation. A previous study showed that the hepatocytes from rats with moderate copper deficiency (liver copper level of 4–8 µg/g dry weight, equal to marginal copper deficiency in our study) [48] have enlarged, bizarre-shaped mitochondria and disarranged endoplasmic reticulum (ER) as assessed by electron microscopy [156]. In rats with severe copper deficiency (liver copper level of ˂2 µg/g dry weight), the hepatocyte ultrastructure displayed dramatic changes characterized by the giant, misshapen mitochondria which occupy most of the cytoplasmic space and squeeze out and obscure otherwise normal-looking organelles. The mitochondrial matrix is less dense than normal [156]. These apparent morphological alterations of mitochondria appear to be linked to their abnormal functions. Whether and how severe copper deficiency affects ER and lysosome function leading to ER stress and defective autophagy remain elusive.

Pharmacological suppression of systemic copper levels with a chelating drug impaired mitochondrial energy metabolism and decreased ATP levels despite induction of glycolysis [157]. Of note, it is well documented that fructose metabolism also leads to ATP depletion [102,105,106]. Thus, one may postulate the additive or synergistic effect of copper deficiency and high fructose intake could be lethal. In fact, this effect has been demonstrated in experimental animals [158,159]. However, the effects of severe copper deficiency in rats fed with fructose can be reversed by replacing fructose with either glucose or starch [50,71].

Fructose and glucose are distinct in several aspects, including intestinal absorption, metabolic pathways and the organ of its major metabolism. The unique features of fructose absorption and metabolism provide clues for mechanisms of copper-fructose interactions. Rats treated with allopurinol, a competitive inhibitor of xanthine oxidase, displayed improved symptoms induced by copper deficient and high fructose diet, including anemia and decreased mortality, and this was associated with a dramatic reduction of uric acid. The beneficial role of allopurinol is likely attributable to protection against the catabolism of purines and increased nucleotides pool [160], suggesting the complexity of copper-fructose interactions in NAFLD.

Copper is required for the activity of COX, and copper deficiency was associated with decreased COX activity in multiple organs, including heart, liver, intestine, in mouse and rat models [124,154,161]. Whether or not copper deficiency induced metabolic phenotypic alteration through COX deficiency mediated mitochondrial dysfunction and the potential molecular mechanisms are not clear. COX (or complex IV) is the terminal enzyme of the electron transport chain in the inner mitochondrial membrane and catalyzes the transfer of electrons from reduced cytochrome c to molecular oxygen. Complex IV is composed of 14 subunits, and three of these (subunits I–III) form the highly conserved catalytic core of the enzyme encoded by mitochondrial DNA. The remaining less conserved subunits are encoded by nuclear genomes and were considered to be related to structural stability and enzyme activity. Highly conserved domains within subunit I include two heme moieties (heme a and a3) and a copper binding site (CuB), and subunit II also contains a copper binding site (CuA). The assembly of the complex IV protein is achieved by more than 20 different assembly proteins [162,163]. *SCO2* encodes a copper chaperone required for the insertion of copper into the active site of subunit II of complex IV, but it is not essential for complete holoenzyme formation. *SCO2* deficient mice exhibit increased adiposity, hepatic steatosis and insulin resistance along with 20%–60% reduction in complex IV activity [164]. In vitro research in human myoblasts demonstrated that COX deficiency due to mutations in *SCO2* can be rescued by copper supplementation [165]. Whether the copper deficiency-induced reduction in COX activity is through SCO2 remains to be elucidated. However, mice lacking SURF1, a complex IV assembly protein, displayed an improved metabolic phenotype, including reduced adiposity, increased insulin sensitivity, and mitochondrial biogenesis despite of more than 50% reduction in COX activity [166,167,168].

In addition, previous studies from ATP7B knockout mice revealed that copper accumulation dysregulated nuclear receptors which contribute to liver function and lipid metabolism, such liver X receptor (LXR), farnesoid X receptor (FXR), retinoid X receptor (RXR), and small heterodimer partner (SHP) [134,169]. However, the effects of copper deficiency and high fructose on the regulation of nuclear receptors remain elusive.

### 6.2. Copper-Fructose Interaction and Hyperlipidemia

Copper-fructose interactions-induced hypercholesterolemia and hypertriglyceridemia have been well demonstrated [48,50,72,142,147,148,170,171,172,173]. In a population-based cohort study with 1197 subjects, dietary copper intake was inversely associated with plasma total cholesterol and LDL-cholesterol. Serum copper levels from a randomly identified subgroup of 231 men were also inversely associated with plasma total cholesterol and LDL-cholesterol, implying a crucial role of copper in cholesterol metabolism [170]. Rats exposed to a copper deficient diet for 3–4 weeks developed hypercholesterolemia, and this effect was more significant when the diet carbohydrate component was solely fructose, but not the starch, suggesting that a copper-fructose interaction is instrumental in the development of hypercholesterolemia. Moreover, hypercholesterolemia is further worsened by a diet high in saturated fat, but not polyunsaturated fat. However, copper-fructose interaction induced hypertriglyceridemia can be exacerbated by both high saturated fat diet and high polyunsaturated fat diet [72,171,173]. Of note, both hypertriglyceridemia and hypercholesterolemia are associated with hepatic iron overload and are ameliorated by dietary iron restriction [147,148]. Restriction of dietary iron intake significantly decreased blood cholesterol and triglyceride levels associated with decreased lipid peroxidation in rats fed with a copper deficient and high fructose diet. Similarly, the severity of copper deficiency was attenuated by the iron chelator, deferoxamine [147,150,174]. Moreover, increased iron intake further increased blood cholesterol and triglyceride levels in copper deficient diet fed rats [148]. Copper deficiency induced hypercholesterolemia is likely due to increased cholesterol synthesis [172]. Hepatocytes isolated from rats fed with a copper deficient diet for 7-8 weeks exhibited 90% reduction of copper content compared to those from adequate copper fed rats. After three hours incubation, these cells displayed 2–3 fold increase in the intracellular glutathione (GSH) synthesis rate along with the increased intracellular and extracellular GSH [175]. Treatment with L-buthionine sulfoximine (BSO), a specific GSH synthesis inhibitor, abolished the hypercholesterolemia and increased HMG-CoA reductase (HMGCR) activity in rats fed with copper deficient diet [142]. These results suggest that copper deficiency induced hypercholesterolemia and increased HMG-CoA are the consequence of increased GSH synthesis. Moreover, the induction of FAS expression was also prevented by BSO in copper deficient rats [144]. One hypothesized mechanism for the increased synthesis of GSH is a compensatory mechanism to the decreased antioxidant defenses due to the decreased cuproenzymes [48,154].

### 6.3. Copper-Fructose Interaction and Glucose Tolerance

Copper-deficient rats displayed improved glucose tolerance when they were switched from high fructose diet to high glucose diet for four weeks after being fed with a high fructose diet for five weeks. Similarly, changing the dietary carbohydrates in the copper-deficient diet from fructose to starch increased insulin levels and decreased blood glucose in response to a glucose tolerance test compared to rats continuously fed fructose. These results suggest that the copper-fructose interaction was more diabetogenic compared to copper-glucose. [71]. In addition, a copper deficient or a marginally copper deficient diet induced impaired glucose tolerance compared to an adequate copper diet, suggesting that copper deficiency may interfere with glucose utilization [176,177].

### 6.4. Copper-Fructose Interaction and Gut Permeability

Our recent study demonstrated that expression of the tight junction proteins, claudin-1 and occludin, was significantly downregulated in the ileum of rats fed with marginal copper deficient diet. This effect was synergistically or additively enhanced by high fructose feeding, suggesting copper-fructose interaction in the small intestine may play a vital role in gut barrier function [47]. A recent study showed that the metabolism of microbiota-derived butyrate in the gut epithelial cells through β-oxidation results in the depletion of oxygen and contributes to the maintenance of “physiologic hypoxia”, which, in turn, leads to the stabilization of HIF-1 [178]. HIF-1 is a transcription factor which plays a central role in the protection of gut barrier function in multiple ways, including transcriptional regulation of tight junction protein expression [179,180], induction of T_regs_ activation [181], and differentiation via transcriptional regulation of FoxP3 [182]. It is known that copper is required for the activation of HIF-1 [119,183]. Our previous study demonstrated that the fecal short chain fatty acid (SCFA), butyrate, was significantly decreased in high fructose fed male rats [184]. However, questions of whether or not decreased fecal SCFAs play a causal role and whether or not copper-fructose interaction induced gut barrier dysfunction is mediated by HIF-1 remain to be elucidated. In addition, in vitro studies demonstrated that increased copper concentration in the culture medium may induce Caco-2 cell apoptosis and increased permeability of the Caco-2 cell monolayer [185,186,187]. Collectively, copper homeostasis plays a crucial role in maintaining intestinal integrity.

### 6.5. Copper-Fructose Interaction and Gut Microbiome

Our data showed that the gut microbiome of rats fed with 30% fructose (*w*/*v*) in the drinking water and AIN-76 based rodent diet (*ad libitum*) for four weeks exhibited an obesity phenotype characterized by a markedly increased ratio of Firmicutes/Bacteroides, and this effect was further exacerbated with a marginal copper deficient diet, associated with increased gut permeability, exacerbated hepatic steatosis and liver injury [47,48]. These findings indicate that copper-fructose interaction may alter the gut microbiome. The mechanisms involved are not clear. Several lines of evidence indicate that copper might be involved in the regulation of gut microbiota and gut barrier function. First, copper has been used as an antimicrobial agent throughout the ages [188], and the response to copper stress varies among different bacteria species [189,190]. Second, one of the copper containing enzymes, diamine oxidase, was found in high concentrations in intestinal mucosa and its circulating enzyme activity serves as a marker of mucosal maturation and integrity, as does the copper level [191,192,193]. Thus, decreased copper levels may exacerbate dietary fructose-induced gut microbiota dysbiosis and/or gut barrier dysfunction. Whether copper-fructose interaction induced gut barrier dysfunction is the direct role of copper-fructose interaction in the intestine and/or mediated by the gut microbiota requires further investigation.

### 6.6. Sex Difference in the Copper-Fructose Interaction

Sex differences in the metabolic effects of fructose and/or copper deficiency have long been noted in the animal studies [158,194,195,196] as well as in humans [197,198], with males being sensitive to the deleterious effects of fructose and/or copper deficiency, and females being protected, which is consistent with the sex differences in NAFLD prevalence [75,199]. However, the mechanistic link between fructose, copper deficiency and sex is not well established. Experimental study from rats implies the level of testosterone in the males may play a role in the severity of copper deficiency [195]. In line with this, evidence from a murine study demonstrated that testosterone robustly suppressed hepcidin transcription through epidermal growth factor receptor (Egfr) signaling, and these suppressive effects were more obvious in male mice than in female mice [200]. Our work also showed significantly decreased plasma hepcidin levels in CuMF male rats compared to controls [141]. Sex differences in the copper-fructose interaction were also shown with regard to the enzyme activities involved in fructose metabolism pathway and their relevant metabolites [201,202]. A previous study showed that female rats displayed lower hepatic KHK and triose kinase activities, but higher triose phosphate isomerase activity compared to male rats in response to high fructose with either adequate copper or copper deficient diet [201]. Moreover, F1P levels were elevated to a greater extent in male rats compared to female rats on copper deficient diet [202]. In addition, high fructose feeding resulted in markedly elevated serum uric acid levels in male rats compared to female rats, and it was further increased by copper deficient diet compared to adequate copper diet [201]. However, inhibition of uric acid generation with allopurinol showed beneficial effects on copper-fructose interactions [160]. Recent studies pointed out that sexual dimorphism in glycerol metabolism and aquaglyceroporins (AQPs) contribute to the lower prevalence of NAFLD in premenopausal women as well as in rodents. However, whether these mechanisms contribute to the sex difference in the copper-fructose interactions remain to be determined (reviewed by Rodriguez et al.) [203]. Collectively, a sex difference in copper-fructose interactions likely contributes to sex variances in fructose metabolism and susceptibility to NAFLD/metabolic syndrome.

## 7. Conclusions

High fructose consumption and low copper availability are two risk factors identified in NAFLD patients. Evidence of copper-fructose interactions comes largely from animal studies. Hepatic iron overload and mitochondrial dysfunction are two important mechanisms. The causal role of high fructose consumption on the impaired copper status in humans as well as copper-fructose interactions in the pathogenesis of NAFLD patients remain to be firmly established. Therefore, larger cohort studies are needed to examine the correlation between copper status and fructose consumption in healthy controls, obese and NAFLD patients. However, we suggest that there are multiple NAFLD phenotypes, with one such NAFLD phenotype being relatively young males with high sugar sweetened beverage (and high fructose) consumption and modestly depressed serum copper/ceruloplasmin. A beneficial role for restricting dietary fructose intake to improve obesity and the metabolic syndrome has been clearly demonstrated and further studies may confirm the additional role of low copper availability.

## Figures and Tables

**Figure 1 nutrients-10-01815-f001:**
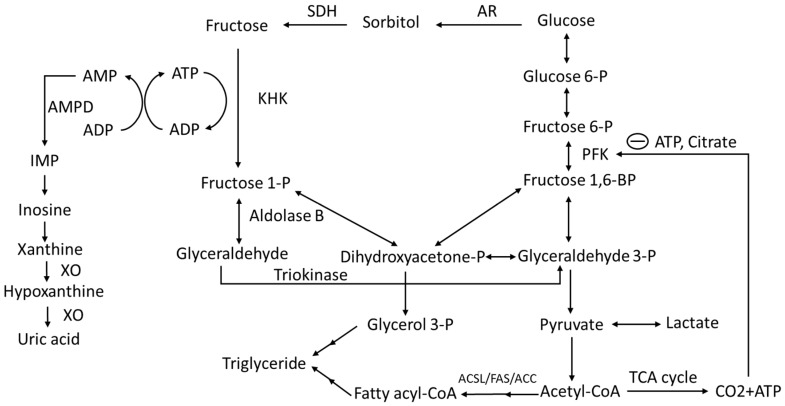
Fructose and Glucose metabolism. AR, aldose reductase; SDH, sorbitol dehydrogenase; KHK, ketohexokinase; PFK, phosphofructokinase; AMPD, adenosine monophosphate deaminase; IMP, inosine monophosphate; XO, xanthine oxidase; ACC, acetyl-CoA carboxylase; FAS, fatty acid synthase; ACSL, long chain acyl-CoA synthetase.

**Figure 2 nutrients-10-01815-f002:**
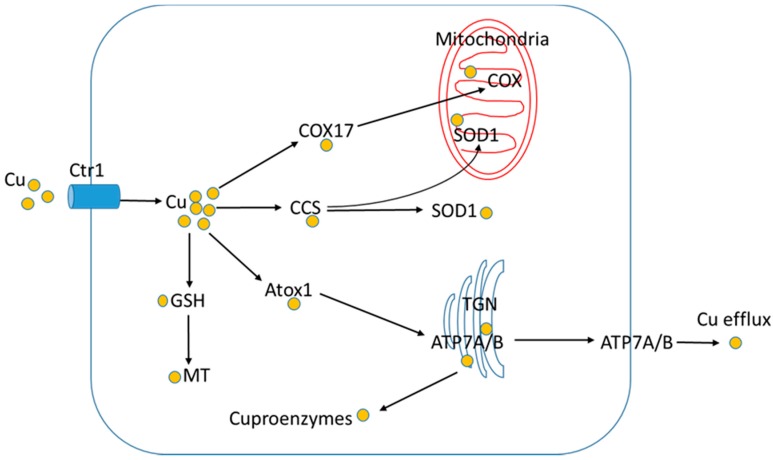
Cellular copper distribution. Ctr1, copper transporter 1; MT, metallothionein; GSH, glutathione; CCS, copper chaperone for SOD1; COX, cytochrome c oxidase; Atox1, antioxidant protein 1.

**Figure 3 nutrients-10-01815-f003:**
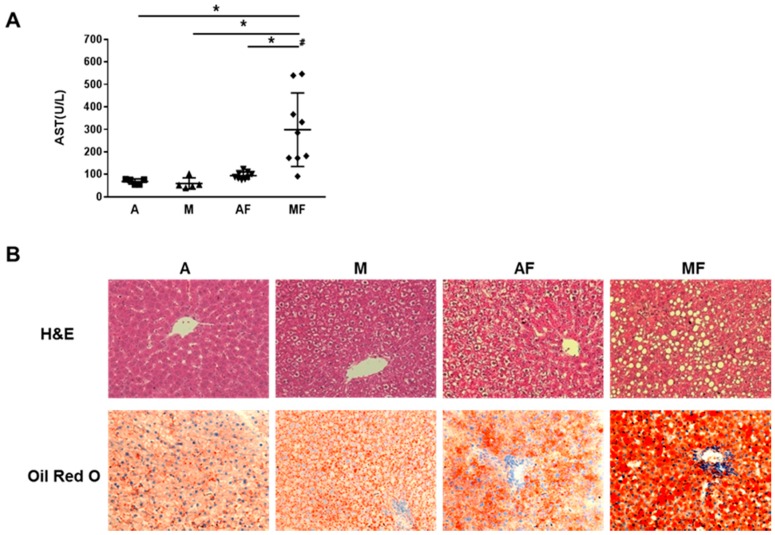
Effects of marginal copper deficiency and fructose feeding on liver injury and lipid accumulation in male weanling Sprague-Dawley rats. (**A**) Plasma AST. (**B**) Representative photomicrographs of the H&E and Oil Red O staining of liver section (200×). Data represent means ± SD (*n* = 5–9) and analyzed by two-way ANOVA, * *p* < 0.05; #, interaction between copper and fructose is significant (*p* < 0.05). AST, aspartate aminotransferase; A, adequate copper diet; M, marginal copper deficient diet; AF, adequate copper diet + 30% fructose drinking; MF, marginal copper deficient diet + 30% fructose drinking.

**Figure 4 nutrients-10-01815-f004:**
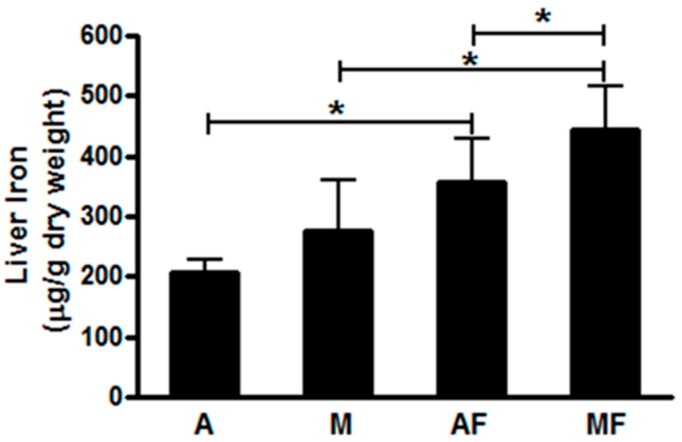
Effect of fructose feeding on liver iron in male weanling Sprague-Dawley rats. Data represent means ± SD (*n* = 5–10) and analyzed by two-way ANOVA, * *p* < 0.05; A, adequate copper diet; M, marginal copper deficient diet; AF, adequate copper diet + 3% fructose drinking; MF, marginal copper deficient diet + 3% fructose drinking.

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
