# Peer review of "Copper-Fructose Interactions: A Novel Mechanism in the Pathogenesis of NAFLD"

_nutrients, 2018, doi:10.3390/nu10111815_

Round 1
Reviewer 1 Report
The review in question addresses the critical role played by dietary fructose in the epidemic of nonalcoholic fatty liver disease (NAFLD), a worrisome complex disease often associated to obesity and T2DM and a typical manifestation of metabolic syndrome. The authors describe the recent advances and research trends in attempting to explain in why and how high fructose consumption and low copper availability are considered serious risk factors predisposing to NAFLD. While hepatic iron overload and mitochondrial dysfunction are described as being the likely mechanisms with which high consumption of dietary fructose leads to NAFLD the authors state that the causal role of copper-fructose interactions in the pathogenesis of NAFLD patients remains a matter that needs to be firmly established. The Authors also review the existence of multiple NAFLD phenotypes and how fructose consumption may impact the disease. Literature is cited suggesting that decreased copper levels may exacerbate dietary fructose-induced gut microbiota dysbiosis and/or gut barrier dysfunction.
I enjoyed reading the manuscript and feel the review in question a valuable and updated source of information for people working in or approaching the field.
The word “Sorbital” reported in Figure 1 should be corrected to “Sorbitol”.
Being glycerol one of the metabolic intermediates of fructose and glucose metabolism involved in fatty liver disease (Figure 1) and existing sexual dimorphism towards the meaning of such polyol in NAFLD the authors may wish to cite a recent review addressing such a matter (Rodriguez et al, Front Endocrinol 2015 – PMID 26594198).
Author Response
The word “Sorbital” reported in Figure 1 should be corrected to “Sorbitol”.
Response: “Sorbital” in Figure 1 has been corrected to “Sorbitol”.
Being glycerol one of the metabolic intermediates of fructose and glucose metabolism involved in fatty liver disease (Figure 1) and existing sexual dimorphism towards the meaning of such polyol in NAFLD the authors may wish to cite a recent review addressing such a matter (Rodriguez et al, Front Endocrinol 2015 – PMID 26594198).
Response: We thank this reviewer for the helpful comment. This article has been cited and highlighted in section 6.6. Sex Difference in the Copper-fructose Interaction.
Reviewer 2 Report
The article titled: Copper-Fructose Interactions: A Novel Mechanism in the Pathogenesis of NAFLD Ming Song, Miriam B. Vos and Craig J. McClai, is a good review very detailed and full of many insights.
Anyway there are some parts that can be improved:
1. There is a long paragraph on absorption, metabolism and metabolic fate of the Fructose: please reduce it.
1.1. On the contrary, the relationship between fructose and NAFLD is missing: the authors should consider inserting some information about it
2. Why the authors didn't consider some of the newest publications on the field? There are more recent publications on the relationship between copper and NAFLD, perhaps the authors have not seen the most recent of 2018, but it is strange that they did not consider those of 2017.There are more recent publications on the relationship between copper and NAFLD, perhaps the authors have not seen the most recent of 2018, but it is strange that they did not consider those of 2017.
Examples are:
- the well linked “Non-Alcoholic Fatty Liver Disease and Nutritional Implications: Special Focus on Copper” by Antonucci L. et al 2017;
- “The role of insufficient copper in lipid synthesis and fatty-liver disease” by Morrell A et al 2017
- “Copper/MYC/CTR1 interplay: a dangerous relationship in hepatocellular carcinoma” by Porcu C. et al 2018;
- “Low Hepatic Tissue Copper in Pediatric Nonalcoholic Fatty Liver Disease” by Mendoza M;
- “Low hair copper concentration is related to a high risk of nonalcoholic fatty liver disease in adults” by Lee SH et al 2018
The authors should consider them and evaluate if introduce them into the manuscript.
In "Conclusions" the line 450 should be written more clearly.
Author Response
1. There is a long paragraph on absorption, metabolism and metabolic fate of the Fructose: please reduce it.
1.1. On the contrary, the relationship between fructose and NAFLD is missing: the authors should consider inserting some information about it
Response: We agree. The section on fructose absorption, metabolism and metabolic fate has been reduced and some information on the relationship between fructose and NAFLD has been added as suggested.
2. Why the authors didn't consider some of the newest publications on the field? There are more recent publications on the relationship between copper and NAFLD, perhaps the authors have not seen the most recent of 2018, but it is strange that they did not consider those of 2017.
Examples are:
- the well linked “Non-Alcoholic Fatty Liver Disease and Nutritional Implications: Special Focus on Copper” by Antonucci L. et al 2017;
- “The role of insufficient copper in lipid synthesis and fatty-liver disease” by Morrell A et al 2017
- “Copper/MYC/CTR1 interplay: a dangerous relationship in hepatocellular carcinoma” by Porcu C. et al 2018;
- “Low Hepatic Tissue Copper in Pediatric Nonalcoholic Fatty Liver Disease” by Mendoza M;
- “Low hair copper concentration is related to a high risk of nonalcoholic fatty liver disease in adults” by Lee SH et al 2018
The authors should consider them and evaluate if introduce them into the manuscript.
Response: One of the articles, “Low Hepatic Tissue Copper in Pediatric Nonalcoholic Fatty Liver Disease” by Mendoza M, has already been cited (reference 54) in our first submission. Actually, this work is from our group. Reference 55 in the first submission, “Low hepatic copper content and PNPLA3 polymorphism in non-alcoholic fatty liver disease in patients without metabolic syndrome. J Trace Elem Med Biol 2017;39:100-107.” by Stattermayer et al. 2017, has also been cited. We have noted the two recent published review articles by Antonucci L. et al. 2017 and Morrell A et al. 2017. Both of them addressed the relationship between copper and NAFLD, which are partially overlapped with our topic-“copper-fructose interactions”. We have added them in our review. We have also added the two articles by Porcu C. et al 2018 and Lee SH et al. 2018 to section 5. Copper Homeostasis and NAFLD. In addition, several recent published articles have been added, including by Heffern et al. 2016 (end of section 2. Epidemiology of NAFLD, fructose consumption, and dietary copper intake), Fujii et al. 2017 (close to the end of Introduction).
In "Conclusions" the line 450 should be written more clearly.
Response: We have rewritten the conclusions.
High fructose consumption and low copper availability are two risk factors identified in NAFLD patients. Evidence of copper-fructose interactions comes largely from animal studies. Hepatic iron overload and mitochondrial dysfunction are two important mechanisms. The causal role of high fructose consumption on the impaired copper status in humans as well as copper-fructose interactions in the pathogenesis of NAFLD patients remain to be firmly established. Therefore, larger cohort studies are needed to examine the correlation between copper status and fructose consumption in healthy controls, obese and NAFLD patients. However, we suggest that there are multiple NAFLD phenotypes, with one such NAFLD phenotype being relatively young males with high sugar sweetened beverage (and high fructose) consumption and modestly depressed serum copper/ceruloplasmin. A beneficial role for restricting dietary fructose intake to improve obesity and the metabolic syndrome has been clearly demonstrated and further studies may confirm the additional role of low copper availability.
Reviewer 3 Report
This is an extensive review of the fructose-copper connection for NAFLD. The senior authors are experts in the field of NAFLD and as such the review gives and excellent overview of the field. The paper is well written, easy to understand and interesting to read. A few minor criticisms.
The description of intracellular Cu metabolism is somewhat dated (2009) since then significant progress has been made, Ctr2 was discovered and described for example. This review unfortunately also overlooked work by the Burkhead lab from the past two years with respect to the fructose-copper connection. In fact Morell et al. published a similar, less extensive, review article about Cu deficiency and increased lipid synthesis.
Lastly, the title of the review article indicated that a novel mechanism for the pathogenesis of NAFLD would be presented. A few (known) mechanisms were discussed but always with a caveat that they are more or less hypotheses at this point.
Author Response
The description of intracellular Cu metabolism is somewhat dated (2009) since then significant progress has been made, Ctr2 was discovered and described for example.
Response: We agree and have added new literature related to copper transporters in section 4. Copper absorption, distribution and utilization.
This review unfortunately also overlooked work by the Burkhead lab from the past two years with respect to the fructose-copper connection. In fact Morell et al. published a similar, less extensive, review article about Cu deficiency and increased lipid synthesis.
Response: We agree and have added the work from Dr. Burkhead’s lab to this review (section 6.1. Copper-fructose Interaction and NAFLD). We have also noted the review article published by Morell and Burkhead’s group and cited them (end of section 5. Copper Homeostasis and NAFLD).
Lastly, the title of the review article indicated that a novel mechanism for the pathogenesis of NAFLD would be presented. A few (known) mechanisms were discussed but always with a caveat that they are more or less hypotheses at this point.
Response: Copper-Fructose Interaction is a conceptual novel mechanism. The likely underlying mechanisms involve hepatic iron overload and mitochondrial dysfunction. However, the cellular and molecular mechanisms are incompletely understood and remain elusive.
Reviewer 4 Report
The present manuscript submitted by Song et al., summarized the significant role played by the copper-fructose interactions during the pathogenesis of the metabolic syndrome with a special emphasis on NAFLD.
NAFLD is the leading cause of chronic liver disease in the United States and other Western countries. The author’s explanation of the various mechanisms of the disease is very lucid. It is a well written and well-structured manuscript. Overall, it is an impressive manuscript and has the potential to grab the attention of wide range of readers.
Author Response
The present manuscript submitted by Song et al., summarized the significant role played by the copper-fructose interactions during the pathogenesis of the metabolic syndrome with a special emphasis on NAFLD.
NAFLD is the leading cause of chronic liver disease in the United States and other Western countries. The author’s explanation of the various mechanisms of the disease is very lucid. It is a well written and well-structured manuscript. Overall, it is an impressive manuscript and has the potential to grab the attention of wide range of readers.
Response: We thank this reviewer for the encouraging comments.